



# Observation based precipitation life cycle analysis of heavy rainfall events in the southeastern Alpine forelands

Stephanie J. Haas[1], Andreas Kvas[1], and Jürgen Fuchsberger[1]

[1]Wegener Center for Climate and Global Change, University of Graz, Graz, Austria

**Correspondence:** Stephanie J. Haas (stephanie.haas@uni-graz.at)

**Abstract.**

Heavy thunderstorms are a typical weather phenomenon during summer in southeast Austria. These fast-developing high-impact rainfall events often result in serious damage and are hard to predict. A profound understanding of the life cycle of these events, from formation to dissipation, is therefore crucial to increase resilience and improve forecasting skills. High-resolution observation datasets, like the one of the WegenerNet 3D Open-Air Laboratory for Climate Change Research (WEGN3D) in Feldbach (Austria), provide unique insights and are especially well suited for these important use cases. Consisting of 156 ground stations, an X-band radar, two radiometers, and 6 global navigation satellite system (GNSS) stations, the WEGN3D delivers highly resolved data, in both space and time, of key atmospheric parameters that enable a detailed investigation of small-scale weather phenomena, such as heavy rainfall events. Here we follow the different stages of the life cycle of 94 heavy rainfall events by investigating multiple atmospheric parameters in WEGN3D and global reanalysis data. Beginning with the event formation stage (i.e., the 8 h before the event), where temperatures are usually already quite high and continue to rise, while the first clouds begin to form, before wind speeds pick up and the sky darkens. Connected to these characteristics, we find an increase in 2 m air temperature anomaly, integrated water vapor (IWV) anomaly, liquid water path (LWP), convective available potential energy (CAPE), and wind speed. Also, a decrease of cloud base height (CBH) can be observed, in accordance with the deepening of the convective cloud system. During the precipitation stage, we find an increase in the spatial variability of precipitation amount, temperature, LWP, and cloud cover, which represents the highly localized character of these events. After a few minutes to hours of intense rainfall, the event is over and has reached the dissipation stage. The parameters that increased during the event formation stage experience a drop during this last stage (i.e., the 16 h after the event), while CBH again reaches its pre-event levels. Our study gives insights into the physical processes connected to the life cycle of heavy rainfall events, by using the WEGN3D's distinct capability to capture characteristic features of such small-scale events, which illustrates the dataset's high potential for improving and verifying weather and climate models.

## 1 Introduction

This study focuses on the life cycle of heavy precipitation events (HPEs) in observation data. These short and highly localized (convective) events strongly influence warm season rainfall in mid-latitude Europe (Taszarek et al., 2019; Lombardo and Bitting, 2024), are particularly hazardous, and often result in serious damage and socioeconomic losses (Schroeer and Tye,





2019; Mateos et al., 2023). Considering that heavy rainfall is projected to increase in both frequency and magnitude (Haslinger et al., 2025; IPCC, 2021), the threat posed by HPEs will not diminish over time. Not only will HPEs themselves pose an even higher risk in our everyday lives, also the number of flash floods and landslides triggered by these events will likely increase (IPCC, 2012; Gariano and Guzzetti, 2016) . In addition to the risks associated with these highly localized events, HPEs are

hard to predict and numerical weather prediction (NWP) models often fail to do so accurately (Sun et al., 2014). A better understanding of HPEs including their evolution in different atmospheric parameters is crucial for the improvement of NWP models and their skill to predict such high-impact events, especially for regions prone to heavy (convective) precipitation.

One of these regions is the Feldbach region (FBR, centered around 46.938°N, 15.908°E) in the southeast of Austria, which we use here as study area. Located at the interface between Alpine and Mediterranean climate, the hilly landscape is character-

ized by hot summers that provide one of the key ingredients of HPEs - high temperatures, resulting in the region's proneness to heavy rains from thunderstorms during that season (Kirchengast et al., 2014; Kabas et al., 2011). This is a characteristic feature of the larger southeastern Alpine forelands, for which the FBR is also representative in terms of (seasonal) precipitation amount (Lombardo and Bitting, 2024). Besides its interesting climatic features, the region is also very well observed since the implementation of the WegenerNet climate station network (WEGN) Feldbach region in 2007, which covers an area of 22 km

× 16 km and provides high-resolution meteorological surface data of the region (Kirchengast et al., 2014; Fuchsberger et al., 2021). The observational data record in the region was broadened in 2020, by the transition of the WegenerNet FBR into the WegenerNet 3D Open-Air Laboratory for Climate Change Research (WEGN3D Open-Air Lab, WEGN3D) through the addition of atmospheric sounding capabilities, consisting of an X-band dual-polarization Doppler precipitation radar, a broadband infrared cloud structure radiometer, a combined microwave/infrared tropospheric sounding radiometer, and a six-station water

vapor sounding Global Navigation Satellite System (GNSS) network.

A common issue in heavy precipitation research is the lack of a single ideal dataset for the analysis of HPE changes over time. While reanalysis datasets like ERA5 (Hersbach et al., 2020) have the advantage of long data records which enable the investigation of HPEs in the context of climate change, their spatial and temporal resolutions are not sufficient to adequately represent HPEs (Haas et al., 2024). High-resolution observation datasets, like the one from the WEGN3D Open-Air Lab, can

give more detailed insights into HPEs, but are usually only available over the most recent decades. Since this study focuses on the life cycle of HPEs rather than the climate change-induced alterations of such events, using observation data is the obvious choice to answer the following research questions:

  – How do specific atmospheric parameters evolve in the hours before and after heavy precipitation events?

  – What spatial effects does rainfall have on surface parameters and moisture distribution?

To answer these research questions, we investigate 8 different parameters connected to heavy precipitation. The chosen parameters target different aspects of HPEs and their associated atmospheric conditions. As already mentioned above, temperature plays a key role in the formation of heavy (convective) rainfall in our study region. Together with convective available potential energy (CAPE) and wind speed, we get a comprehensive overview of the state of the atmosphere in the time before and after HPEs. The moisture-related parameters absolute humidity (AH), relative humidity (RH), integrated water vapor



**Table 1.** Description of the investigated parameters, incl. unit, data source, and temporal resolution.

| Parameter | Description | Unit | Data source | Temporal resolution |
|---|---|---|---|---|
| PA | Precipitation amount | mm h$^{-1}$ | WEGN grid & Radar | 5 min |
| T2M | 2 m air temperature | K | WEGN grid data | 5 min |
| WindSpeed | Wind speed | m s$^{-1}$ | WEGN station 44 | 5 min |
| RH | Relative humidity profile | % | Radiometer (MW[a]) | 10 min |
| AH | Absolute humidity profile | g m$^{-3}$ | Radiometer (MW) | 10 min |
| T | Tropospheric temperature profile | K | Radiometer (MW) | 10 min |
| IWV | Column-integrated water vapor | mm | GNSS | 10 min |
| LWP | Liquid water path | kg m$^{-2}$ | Radiometer (MW) | 10 min |
| CAPE | Convective available potential energy | J kg$^{-1}$ | Radiometer (MW) | 10 min |
| CBH | Cloud base height | m | Radiometer (MW & IR[b]) | 10 min |
| CloudCover | Regional cloud cover | % | Radiometer (MW & IR) | 10 min |

[a] microwave, [b] infrared

(IWV), and the cloud properties liquid water path (LWP) and cloud base height (CBH) complement this general overview with insights into the rain cells themselves and their evolution before and after such events. The chosen parameters enable us to investigate HPEs in a multifaceted way to gain a better and more profound understanding of these events.

## 2 Data and methods

### 2.1 Data

In this study, we use the latest level 2 (L2) observational data from the WEGN3D Open-Air Lab, consisting of atmospheric sounding data cubes (Kvas et al., 2024) and climate station data (Fuchsberger et al., 2024). These L2 datasets comprise quality-controlled, high-resolution observations of atmospheric state variables from the surface and near-surface to the upper troposphere. The WEGN3D datasets include radiometer-derived time series of CBH and LWP; radiometer-derived tropospheric profiles of air temperature (T), RH, and AH; GNSS-derived IWV; gridded 2 m air temperature (T2M), gridded rain-gauge

and radar-derived precipitation amount (PA), and wind speed (WindSpeed) station time series. We further use all-sky scans of infrared brightness temperatures to detect cloud pixels (e.g., Feister et al., 2010) from which we compute the regional cloud cover (CloudCover). These observed quantities are aggregated to time series with 5 min (T2M, WindSpeed, PA) and 10 min (CBH, LWP, T, RH, AH, IWV, CloudCover) temporal resolution. Table 1 gives an overview of the parameters used in this study, together with their unit, data source and temporal resolution.

The dataset covers the horizontal extent of the study area and has a vertical extent of approximately 250 m to 10 km above mean sea level. T2M is given on a 100 m by 100 m terrain-following grid, which is obtained from the climate station observations by inverse distance weighted interpolation and local temperature lapse rates (Hocking, 2020). The radiometer site and the




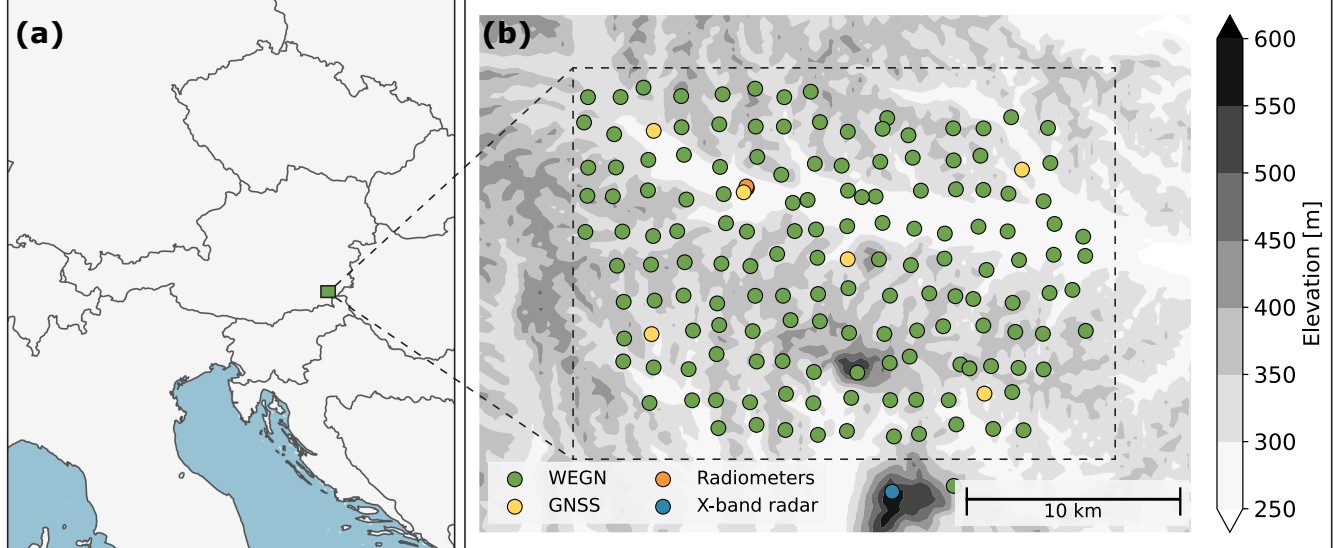

**Figure 1.** Location of the WEGN3D Open-Air Lab (a) and its individual stations (b). The colors of the circles indicate the station type: Green - climate station; orange - radiometer site (microwave/infrared and infrared); yellow - GNSS station; blue - X-band precipitation radar. The elevation (in m) is indicated by the gray colors.

wind speed sensor, where CBH, CloudCover, LWP, T, RH, AH, and WindSpeed are observed, are located approximately at the center of the coverage area. The six-station GNSS network, that provides IWV, is evenly distributed throughout the coverage

area. For spatially distributed data variables (PA, T2M, IWV), we compute the regional spatial mean and the regional spatial variability, represented by the standard deviation, to obtain a single representative time series. Figure 1 gives an overview of the sensor locations and coverage. We also compare the rainfall life cycle using reanalysis data, specifically ERA5 (Hersbach, H. et al., 2023a, b), produced at the European Centre for Medium-Range Weather Forecasts in their $0.25° \times 0.25°$ gridded version. This reanalysis provides the same data variables as the observational datasets used in this study, albeit with a coarser

spatial and temporal resolution.

## 2.2  Methods

### 2.2.1  Case study event selection

Based on the WEGN event database, we select heavy precipitation events (HPEs) for our case study. The predominant type of precipitation in our study region is convective precipitation, which occurs mainly during the months of the warm season (April

to October). We therefore first select all events from these months and discard the ones from the colder months (November to March). A detailed description of the WEGN event type classification is given in Appendix A. Furthermore, we only consider events that occurred later than 2020 to ensure that all of our observational data (climate station network, radiometers, GNSS,





and radar) are available for all selected events. Given our focus on heavy precipitation, we require that our events have a maximum precipitation rate that exceeds the 90$^{th}$ percentile value (roughly

### 2.2.2 Rainfall life cycle analysis

To analyze the life cycle of the HPEs recorded in the WEGN3D, we investigate 10 different parameters connected to heavy precipitation. The chosen parameters are: integrated water vapor (IWV), liquid water path (LWP), convective available potential energy (CAPE), 2 m air temperature (T2M), air temperature profiles (T), cloud base height (CBH), cloud cover (CloudCover), wind speed (WindSpeed), relative humidity (RH), and absolute humidity (AH). Additionally, we inspect the temporal and spatial evolution of PA, T2M, and IWV during the events. Table 1 gives a detailed overview of the parameters. All parameters are directly available in the WEGN3D dataset except for the CAPE index, which we calculate from air pressure profiles, and radiometer-derived T and RH profiles. The preprocessing steps are the same for all analyzed parameters. We select data from all time steps between 8 h before the event and the start of the event, as well as data from all time steps from the end of the event until 16 h after the event ends to investigate the event formation and dissipation.

For parameters which are provided on a spatial grid (T2M and CBH), we calculate the spatial mean value before the temporal selection to get a single time series for each parameter and event. To assess the precipitation life cycle in our observation data, we then calculate the median of all event samples per timestep for each of the above-mentioned parameters. Additionally, the 25$^{th}$ and 75$^{th}$ percentiles of all event samples per timestep are used as an indicator of the parameters' spreads around the event median. In some cases, anomaly time series reveal more about the development of HPEs than the time series of the absolute values. For these parameters (IWV, CBH, T2M, and AH), we first calculate the mean over the investigated time span (8 h before the event to 16 h after the event) and subtract it from the data before calculating the median and spread of the samples.

The data preparation and analysis procedures for ERA5 data are identical to those outlined above for the WEGN data. ERA5 provides all of the investigated parameters except absolute humidity, which we calculate from the pressure, temperature, and relative humidity data.

For the investigation of the precipitation stage (i.e., the part of the life cycle with rainfall), we focus on the maximum precipitation amount, the 2 m air temperature anomaly, and the IWV anomaly, as well as their spatial variability. We normalize the duration of each single event to make them comparable and repeat the data preparation steps described above.

## 3 Results

First, we investigate some general event statistics of the 94 HPEs selected by the methods described in Section 2.2.1 (Fig. 2). HPEs are by nature quite intense and short events, i.e. all of the precipitation accumulates in a short period of time. This is reflected in high maximum precipitation rates of more than 50 mm h$^{-1}$ (Fig. 2a), comparatively low mean precipitation rates below 5 mm h$^{-1}$ (Fig. 2b), and event durations shorter than 4 h (Fig. 2c). We find more HPEs during the afternoon (Fig. 2d) and during the warmer summer months June-August (Fig. 2e). This is due to the close link of convective precipitation to temperature. As mentioned above, the predominant precipitation type in our study region during the warm season is convective.





**Figure 2.** Event statistics of the 94 HPEs selected for the study. (a) Maximum hourly precipitation rate in mm h$^{-1}$, (b) mean hourly precipitation rate in mm h$^{-1}$, (c) event duration in hours, (d) local time of event maximum, (e) month of event occurrence, and (f) precipitation type classified by the WEGN event classification (see Appendix A).

This is also clearly visible in the histograms of the event statistics, where most events are classified as convective or mixed events (Fig. 2f), meaning that they have at least a convective part.

## 3.1 Event formation and dissipation stages

Next, we are interested in the development of multiple parameters that are linked to HPEs from 8 h before to 16 h after the event (Fig. 3). First, we focus on the WEGN3D data (solid colored lines in Fig. 3). Like Wang and Hocke (2022), we find that the median IWV anomaly increases in the hours prior to the event. During the event, the IWV anomaly drops by about 2 mm, followed by a continuous gradual decline during the hours after the event. The LWP is mostly constant throughout the investigated time span with a sharp increase of roughly 0.4 kg m$^{-2}$ in the hour directly before the event onset and a clear drop after the event. Both, the IWV anomaly and the LWP, represent the increase of water (liquid and gaseous) in the atmosphere





before HPEs. After the events, the water precipitated to the surface, and both the IWV anomaly and the LWP have lower levels

than before. Further, we find a clear build up of CAPE, peaking one hour before the event onset, followed by drastic drop of nearly 300 J kg$^{-1}$, underlining the convective character of the investigated events. The time lag between the CAPE peak and the event onset is the result of deep convection, which weakens CAPE through multiple mechanisms. One of these mechanisms is the warming of the upper air layers as a result of the upward transfer of latent heat released during condensation, resulting in a more stable atmosphere. This stabilizing effect is further supported by the cooling of the lower atmosphere through downdrafts

(Barbero et al., 2019). Another effect of the convective nature of HPEs is the deepening of the convective cloud system, represented by a decrease in the CBH anomaly of about 1000 m prior to the event onset. Concerning temperature, we find that the 2 m air temperature anomaly is roughly 5.5 K lower after the event than it was directly before the beginning of the event. The wind speed anomaly exhibits a stark rise (+1.5 m s$^{-1}$) before the event and is mainly constant in the hours following the end of the event.

Figure 3 additionally shows the precipitation life cycle in the ERA5 data (dotted gray lines). ERA5 describes the general tendencies of IWV anomaly, CAPE, and T2M anomaly before and after the event well, despite the lower spatial and temporal resolution of the dataset. This is in line with the expected physical behavior. Cloud and wind related parameters do, however, not show any patterns related to the HPEs.

Using the atmospheric sounding capabilities of the WEGN3D Open-Air Lab, we now investigate the vertical structure of

temperature anomaly, relative humidity, and absolute humidity anomaly before and after the event (Fig. 4a-c). In agreement with the 2 m air temperature anomalies shown above, we find a stark contrast in air temperature of roughly 5.5 K between the time before and after the event. Considering that our events mainly occur in summer in the afternoon (Fig. 2), high temperature anomalies are linked to high absolute temperatures, which are one of the key ingredients of convective precipitation. The high temperature anomalies found in Fig. 4a therefore further underline the convective nature of our events. Another characteristic

of convective precipitation can be observed in Fig. 4b where the vertical structure of relative humidity is shown. Areas with high relative humidity values enable us to make assumptions about the location and extent of the cloud. These areas expand vertically close to the event onset, corresponding to the deepening of the convective cloud system. Using the lower contour of the 80-90 % RH area as a proxy for the CBH, we see a decrease in CBH of about 1 km in the 8 h before the event. This is in line with the drop in CBH anomaly already detected in Fig. 3d. To complement the findings of the vertical structure of

temperature anomaly and relative humidity, we also investigate the absolute humidity anomaly (Fig. 4c). As one might expect, the absolute humidity anomalies are higher (by roughly 2 g m$^{-3}$) in the time before the event than they are in the time after the HPEs occurred. The smaller values of absolute humidity anomalies in the time after the event can be explained by air moisture precipitating to the ground, and also from the fact that the air is colder after the event occurred and can hence hold less moisture than before.

Again, we are also studying the life cycle using the reanalysis dataset. For that purpose, we examine the vertical structure of temperature anomaly, relative humidity, and absolute humidity anomaly in ERA5 data (Fig. 4d-f). The general pattern in temperature anomaly of higher values before the event and lower ones after is also visible in the ERA5 data. We find temperature anomalies in the range of $\pm$ 2 K with a shift from high to low anomalies about 3 h after the event. The sharp drop





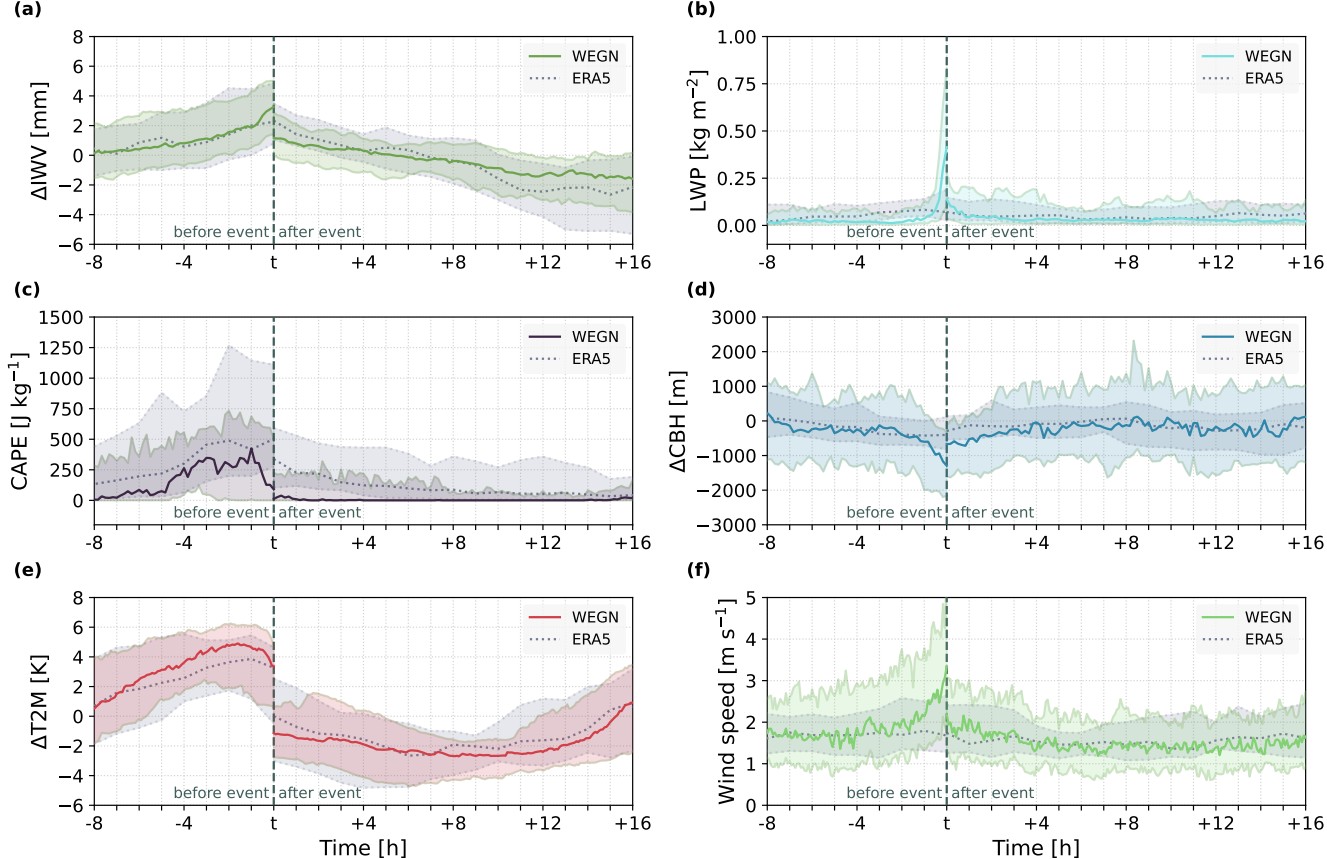

**Figure 3.** Climatology of different HPE parameter from 8 h before to 16 h after the event for WEGN3D data (solid colored lines) and ERA5 (dotted gray lines). The time is given as $t \pm$ hours after/before the event. The dashed line marks the event $t$, since the hours of the event itself (i.e., when rainfall occurred) are not depicted here. The parameters shown are: IWV anomaly (a), LWP (b), CAPE (c), CBH anomaly (d), 2 m air temperature anomaly (e), and wind speed anomaly (f). The lower and upper edges of the shaded corridors correspond to the 25[th] and 75[th] percentiles respectively.

in temperature during the last hour before the event is, however, less pronounced in ERA5 compared to the WEGN3D data.

We attribute this to the coarser time resolution of the reanalysis, which may not be able to accurately capture sudden changes in the pre-rainfall environment. Concerning relative humidity, the HPEs seem to have an effect on ERA5 RH values at higher altitudes of about 5-8 km, where RH increases for a short period after the event before returning to its pre-event levels. The RH can be used to get an idea of the rough locations of the storm cloud. In agreement with the CBH anomalies shown in Fig. 3, the clouds are relatively stable over the whole observed time span, showing no indication of the deepening of the convective

cloud system. Similar to the patterns found in the temperature anomalies, the absolute humidity anomalies also exhibit positive





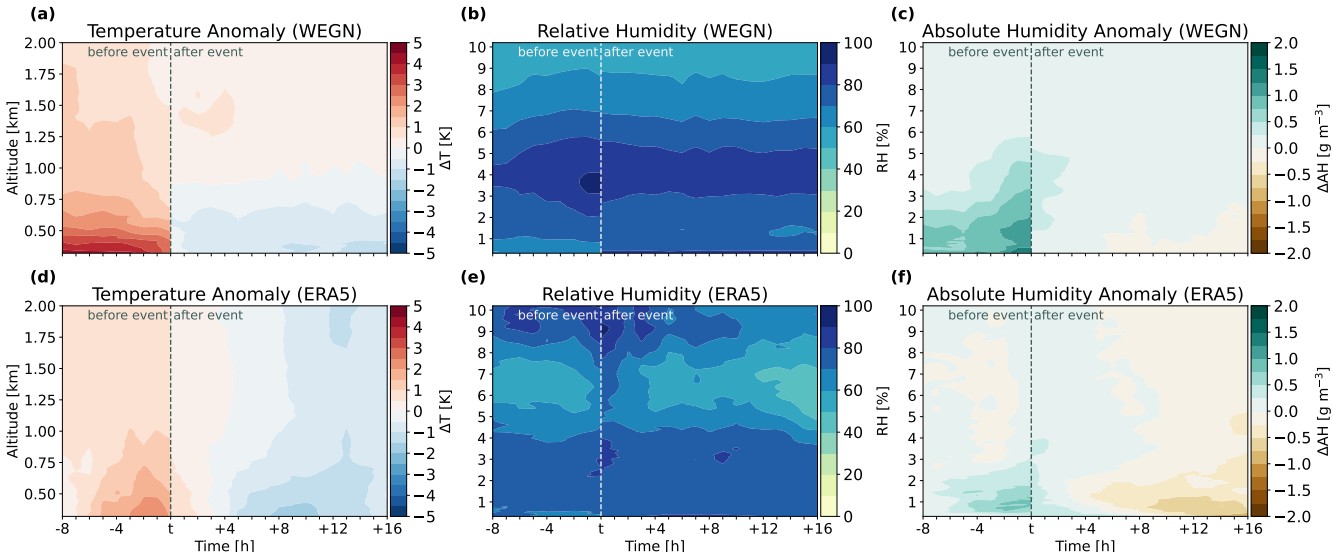

**Figure 4.** Median vertical structures of temperature anomaly (left column), relative humidity (middle column), and absolute humidity anomaly (right column) for WEGN3D (top row) and ERA5 (bottom row) from 8 h before to 16 h after the event. The time is given as $t$ $\pm$ hours before/after the event. The dashed line marks the event $t$. The temperature anomalies are shown for the lower atmosphere up to 2 km, the other parameters up to 10 km.

values up to about 4 h after the event, and negative values afterwards. The elevated AH anomalies are all located below 3 km and have values in the range of $\pm\,1\,\mathrm{g\,m^{-3}}$.

Figure 5a shows that the median regional cloud cover exhibits a large spread in the time span between 8 h and 2 h before the event, but starts to consistently increase towards 100 % at 2 h before the event onset. This increase in cloud cover coincides with the T2M maximum, which also occurs approximately 2 h before the event (see Fig. 3e). Following the T2M maximum, the surface cooling effect of the increased cloud cover, reduced cloud base height, and increased LWP (see Fig. 3d and Fig. 3b) is then clearly visible in the T2M time series (see Fig. 3e). The spatial variability in T2M (Fig. 5b) starts to increase consistently by 0.2 K in the last hour before the event. While this is considerably later than the cloud cover increase, it coincides with the sudden increase in LWP (see Fig. 3b) before the event. This suggests that a thicker cloud layer is required for localized cooling to occur. After the event, we see a substantial drop in T2M variability which indicates that rainfall and persistent cloud cover lead to a homogeneous horizontal temperature distribution.

Figure 5c shows an increase in IWV spatial variability that already starts approximately 2 h before the precipitation stage. This indicates the onset of convection (Weckwerth, 2000). After the event, the IWV variability drops to pre-event levels.



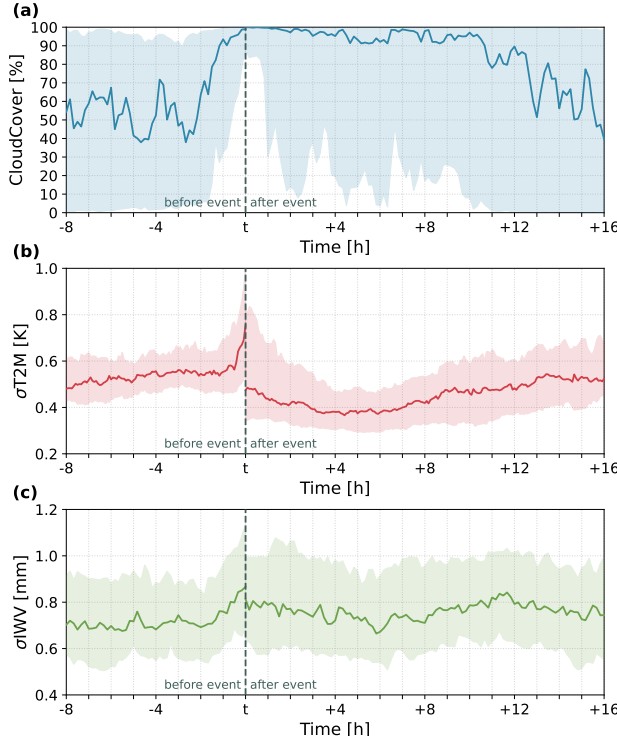

**Figure 5.** (a) Temporal evolution of regional cloud cover (CloudCover) for all events, from 8 h before to 16 h after the event. Median spatial variability of (b) 2 m air temperature ($\sigma$T2M) anomaly and (c) integrated water vapor anomaly ($\sigma$IWV) for all events, from 8 h before to 16 h after the event. The time is given as $t \pm$ hours before/after the event. The dashed line marks the event $t$. The lower and upper edges of the shaded corridors correspond to the 25th and 75th percentiles respectively.

## 3.2 Precipitation stage

The precipitation stage of each event starts with the first recorded occurrence of surface rainfall in the study area and lasts until precipitation ceases. In this event stage, we focus our analysis on the spatial and temporal variability of PA, T2M, and IWV, and their interaction. Profiles of RH, AH, and T, as well as time series CBH and LWP are not available during the precipitation stage, as the microwave radiometer observations used in their retrieval become heavily biased once the instrument is coated with water (Löhnert and Crewell, 2003). Since the duration of the vast majority of events is similar, around 2 h (see Fig. 2),

we normalize their duration to make them comparable. Figure 6 shows a snapshot of the regional T2M anomaly, along with accumulated precipitation amounts derived from radar and gauge measurements. We can see that the spatial pattern of the near-surface cooling is highly correlated with the accumulated precipitation amount, as a direct consequence of the cooler precipitation hitting the ground and subsequent evaporation.

Figure 7 depicts the development of the maximum precipitation amount (PAx), 2 m air temperature anomaly, and IWV

anomaly during the event, as well as, the spatial variability of these parameters. The steady increase in PAx up until about 27 %





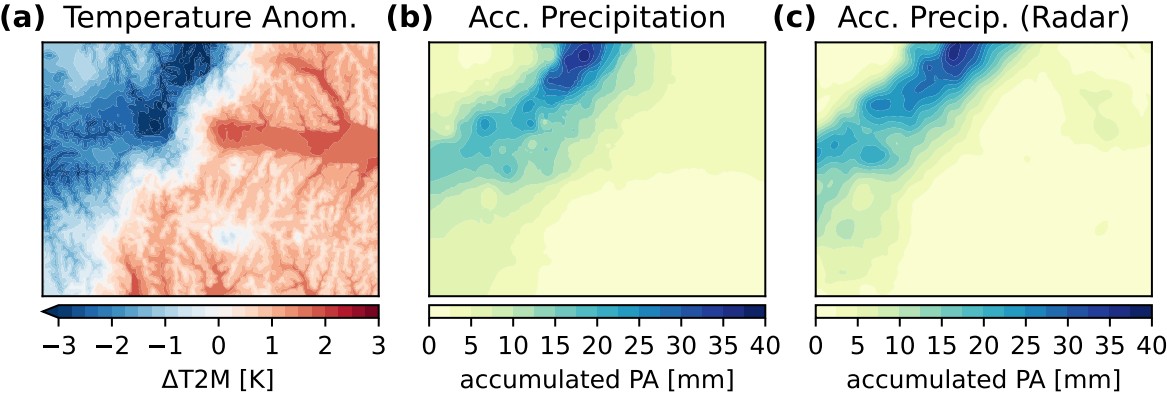

**Figure 6.** Snapshot of the 2 m near surface temperature (T2M) regional anomaly in panel (a) and accumulated precipitation amount from radar and rain gauges in panels (b) and (c), 20 min into a selected event.

into the event corresponds with the rain cell entering the FBR region. A large amount of the investigated HPEs are of mixed precipitation type (Fig. 2f), meaning that the events could consist of a convective event that is directly followed by a stratiform event which is probably the reason for the slower decrease in PAx after the peak. This results in the asymmetric behavior of PAx found in Fig. 7a. Another effect of these stratiform follow-up events is the small spread of PAx towards the end of the events. The spatial variability of precipitation during the event (Fig. 7b) matches the development of PAx described above. Again, the spread here illustrates the highly localized character of the events, as well as the effect that some of the events are displaced by stratiform events towards the end.

As already shown and described above, near-surface cooling is highly correlated with the accumulated precipitation amount (Fig. 7c & d). The T2M anomaly steadily decreases throughout the whole event, resulting in roughly 3 K colder temperatures at the end of the event in comparison to the event onset. For the spatial variability of T2M, the maximum occurs at approximately 15 % into the event which amounts to 18 min for a 2 h rainfall event. At this stage, the study region is only partially affected by precipitation, leading to localized precipitation-induced cooling. Once precipitation occurs in the whole study region, the temperature reduction becomes more uniform, resulting in a reduction of T2M spatial variability. The strong correlation between the spatial patterns of PA and T2M is also evident in the almost identical temporal evolution of the PA spatial variability (Fig. 7b), indicating that the precipitation-induced cooling is also the main driver behind the temporal evolution of the T2M spatial variability shown in Fig. 7f.

Finally, we have a look at the IWV anomaly during the event (Fig. 7e). Similar to the T2M anomaly, the IWV anomaly exhibits a decrease during the time of rainfall. The spatial variability of IWV shown in Fig. 7f also follows a pattern similar to that of T2M, with an increase in variability until 30 % of the event duration, followed by a slow decrease. The maximum of the median spatial IWV variability with 1 mm is substantially higher during the precipitation stage than before and after the event, where we observe a range of 0.7 mm - 0.8 mm. This is a consequence of local cooling in rainfall-affected areas and







**Figure 7.** Median (a) maximum precipitation amount, (b) spatial variability of precipitation amount, (c) 2 m air temperature, (d) spatial variability of T2M, (e) median IWV anomaly, and (f) spatial variability of IWV for all events during the precipitation stage. The lower and upper edges of the shaded corridors correspond to the 25th and 75th percentiles respectively.

the subsequent reduced capacity of the air column to hold water vapor, as well as the sensitivity of GNSS-derived IWV to liquid water in the atmosphere (Solheim et al., 1999). The amount of IWV slowly decreases by about 13 % over the course of the precipitation phase. This contrasts the findings of Wang and Hocke (2022), who only find a minor reduction in IWV
for precipitation events shorter than 8 h. We attribute this different behavior to the fact that the events in our study only take place in the warm season and are primarily of convective nature, which results in higher air temperatures before the events. Combined with the lower altitude of the study region, the absolute IWV we observe is higher, and thus also the potential for a larger IWV reduction is higher.



## 4 Discussion

The life cycle analysis of 94 heavy precipitation events confirmed that these events are mainly of convective nature in our study region, which is characterized by its close link to temperature, short durations, and high intensities. Before we discuss the investigated life cycle of HPEs in observation and reanalysis data, it is important to remember how such HPEs usually unfold: let's assume a typical summer day with morning temperatures that are already well above $20\,°C$. During the day, the temperature further rises and the first clouds begin to form. The sky darkens and is slowly filled with towering cloud formations,

while a gentle breeze transforms into strong winds. Shortly afterwards, the first raindrops begin to fall. Within minutes, the rainfall intensifies drastically and we are in the middle of a full-grown heavy convective precipitation event.

Considering this typical life cycle of HPEs in our study region, we find that the physical processes and effects connected to these events are adequately represented in the WEGN3D data. We clearly detect local precursors of HPEs, such as the rise in temperature, illustrated by the temperature anomaly shown in Figures 3e & 4a. The energy build-up, which is also linked

to the rise in temperature, is reflected by the increase of CAPE in the hours prior to the event onset (Fig. 3c). The drop of the cloud base height anomaly prior to the event (Fig. 3d & 4b) corresponds to the expected deepening of the convective cloud system and the stark rise in wind speed before the event onset is also clearly visible in the WEGN3D data (Fig. 3f). During the precipitation stage, precipitation triggers a cooling effect, which we consider to be the main driver of the temporal evolution of the T2M spatial variability during the time of rainfall. This proves the WEGN3D's capability to capture the characteristics of

high-intensity small-scale rainfall events with unique detail.

Our findings are not only physically plausible, they are also in agreement with previous studies. The observed increase in IWV anomaly before event onset confirms the findings of previous studies such as Wang and Hocke (2022), Benevides et al. (2015), Wang et al. (2024), and Sapucci et al. (2019). Since all of these studies focus on different regions (Wang and Hocke (2022): Swiss Plateau; Benevides et al. (2015): Portugal; Wang et al. (2024): Andalusia, Spain; Sapucci et al. (2019): Brazil;

this study: Austria), the observed rise in IWV (anomaly) seems to be a fundamental feature of heavy (convective) precipitation events. Furthermore, the rapid increase in LWP before HPEs, and the increased levels of RH after a rainfall event, are in agreement with the findings of Wang and Hocke (2022), which gives indications for further characteristic features of HPEs in observation data.

The WEGN3D's capability to adequately capture very characteristic features of high-intensity small-scale rainfall events

while being solely observation-based (i.e., generated without any models) illustrates the high potential for applications of this dataset in the improvement and verification of weather and climate models.

As mentioned in the introduction, our study region is highly affected by convective rainfall events, which are particularly hazardous and often result in serious damage and socioeconomic losses (Schroeer and Tye, 2019). Although the FBR is comparatively small ($22\,km \times 16\,km$), it can still be used as a representative example for the larger surrounding region in terms

of (seasonal) precipitation amount and predisposition to convective precipitation (Lombardo and Bitting, 2024), making the findings of this study also applicable for a larger geographical region. Considering the projected increase of these events in a warming climate (IPCC, 2021), the threat posed by heavy precipitation events will likely increase over time as well. Under-





standing HPEs in great detail and being able to monitor them correctly is therefore a crucial skill, which is especially important for highly affected regions such as the southeastern Alpine forelands and its surroundings. Our study shows that the WEGN3D

dataset can play an important role in this difficult and important task.

## 5 Conclusions

This study investigates the life cycle of 94 heavy precipitation events in observation (WEGN3D) and reanalysis (ERA5) data in the southeastern Alpine forelands, to answer the following research questions:

– How do specific atmospheric parameters evolve in the hours before and after heavy precipitation events?

– What spatial effects does rainfall have on surface parameters and moisture distribution?

With respect to the first research question, we find that in the hours prior to an HPE the IWV anomaly, LWP, CAPE, 2 m air temperature anomaly, and wind speed increase, while the CBH anomaly decreases. In the hours following an HPE all of these parameters drop drastically, except for the CBH anomaly which rises back up to pre-event levels. These findings are well in line with previous studies and the expected physical behavior of such events.

In the hour before the event onset, the spatial T2M variability starts to increase, coinciding with the increase detected in LWP (i.e., the thickening of the cloud system). In addition, the spatial IWV anomaly variability indicates the onset of convection in the 2 h prior to the event. The rainfall itself leads to a localized cooling effect, which is clearly visible in the spatial variability of T2M and IWV anomaly during the event. After the event, a homogeneous horizontal temperature distribution is detectable.

Our study shows that despite being solely based on observations, the WEGN3D is very skilled in monitoring HPEs and their

characteristics, which illustrates the dataset's high potential for highly relevant application in the improvement and verification of weather and climate models.

*Data availability.* The WegenerNet 3D Open-Air Laboratory L2 v1.0 data are available under the Creative Commons Attribution 4.0 International (CC BY 4.0) license on the WegenerNet Data Portal (https://wegenernet.org/portal, last accessed 2025-02-09) at https://doi.org/10.25364/WEGC/WPS3D-L2-10.

The WegenerNet climate station network Level 2 data version 8.0 are available under the CC BY 4.0 license on the WegenerNet Data Portal (https://wegenernet.org/portal, last accessed 2025-02-09) at https://doi.org/10.25364/WEGC/WPS8.0:2024.1.

ERA5 hourly data on single levels from 1940 to present are available through the Copernicus Climate Change Service (C3S) Climate Data Store (CDS), at https://doi.org/10.24381/cds.adbb2d47.

ERA5 hourly data on pressure levels from 1940 to present are available through the Climate Data Store (CDS) of the Copernicus Climate

Change Service (C3S), at https://doi.org/10.24381/cds.bd0915c6.





## Appendix A: Precipitation event classification

To classify HPEs, we make use of 5 min gridded T2M and gridded PA to characterize periods of precipitation within the 22 km $\times$ 16 km study region, in terms of their duration, precipitation amount, precipitation intensity, spatial variability, precipitation-affected area, precipitation phase, and convective potential. A precipitation event is defined as a contiguous sequence of epochs

where the maximum 5 min precipitation amount exceeds 0.19 mm (at least two bucket tips in 5 min), the average 5 min precipitation amount exceeds 0.0039 mm (at least 0.05 mm h$^{-1}$), or where the maximum 5 min precipitation amount exceeds 0.009 mm (at least one bucket tip in 5 min) if the 5 min gridded T2M is below 2 °C. If a time period is identified as a precipitation event, we compute further statistics from the $K$ epochs contained in the event. The statistics include the regional T2M average $T_{\mathrm{A}}$ over all $M$ grid points for the event duration, defined as

$$T_{\mathrm{A}} = \frac{1}{M} \sum_{i=0}^{M-1} \frac{1}{K} \sum_{k=0}^{K-1} T(t_k, \boldsymbol{x}_i), \tag{A1}$$

where $T(t_k, \boldsymbol{x}_i)$ is the 5 min T2M at the grid point $i$ and epoch $k$. Additionally, the fractional area at which precipitation occurred, $F_{\mathrm{PA}}$, is defined as

$$F_{\mathrm{PA}} = \frac{1}{A} \sum_{i \in S} a_i \text{ with}$$

$$S = \left\{ i \left| \left( \sum_{k=0}^{K-1} PA(t_k, \boldsymbol{x}_i) \right) > 0.09 \,\mathrm{mm} \right. \right\}, \tag{A2}$$

where $a_i$ is the area associated with the $i$-th grid point and $A$ is the total area covered by the grid data product. We further compute the maximum fractional area covered by precipitation $F_{\mathrm{PM}}$ on at least one epoch with

$$F_{\mathrm{PM}} = \max \left\{ \frac{A_k}{A} \middle| k \in \{0, \ldots, K-1\} \right\} \text{ with}$$

$$A_k = \sum_{i \in S_k} a_i \text{ and } S_k = \{i | PA(t_k, \boldsymbol{x}_i) > 0.09 \,\mathrm{mm}\}, \tag{A3}$$

and the normalized spatial variability $N_{\mathrm{SV}}$ defined as

$$N_{\mathrm{SV}} = \frac{1}{\bar{\bar{P}}_{\mathrm{R}}} \frac{1}{M-1} \sum_{i=0}^{M-1} \left( \bar{P}_{\mathrm{R}}(\boldsymbol{x}_i) - \bar{\bar{P}}_{\mathrm{R}} \right)^2, \tag{A4}$$

where $\bar{P}_{\mathrm{R}}(\boldsymbol{x}_i) = \frac{1}{K} \sum_{k=0}^{K-1} PA(t_k, \boldsymbol{x}_i)$, and

$$\bar{\bar{P}}_{\mathrm{R}} = \frac{1}{M} \sum_{i=0}^{M-1} \frac{1}{K} \sum_{k=0}^{K-1} PA(t_k, \boldsymbol{x}_i).$$

The maximum precipitation sum $P_{\mathrm{SM}}$ is defined as

$$P_{\mathrm{SM}} = \max \left\{ \sum_{k=0}^{K-1} PA(t_k, \boldsymbol{x}_i) \middle| i \in \{0, \ldots, M-1\} \right\}. \tag{A5}$$



**Table A1.** Parameters used for the event classification algorithm.

| Parameter | Description |
|---|---|
| $T_A$ | regional T2M average |
| $P_{SM}$ | maximum precipitation sum |
| $N_{SV}$ | normalized spatial variability |
| $F_{PM}$ | maximum fractional area covered by precipitation |
| $D$ | event duration |
| $F_{PA}$ | fractional precipitation area |

Together with the event duration $D$, these features allow us to further divide precipitation events into the classes convective, stratiform, mixed, weak, and snow. Convective-class events are characterized by high precipitation intensity, high spatial variability, small spatial extent, and a high-enough surface temperature to allow for convection. Stratiform-class events exhibit low spatial variability, large spatial extent, longer duration, and lower surface temperatures. An event is classified as mixed when it cannot be unambiguously identified as convective or stratiform. This typically occurs when convective precipitation is

followed by prolonged rainfall with decreasing intensity. The remaining two classes, which are not considered in this study, are snow which can only occur when the surface temperature is close to freezing level and weak with very low mean precipitation amount.

These class definitions are realized in a decision tree structure, which is applied to the grid-derived features in daily chunks from 6:00 to 6:00 UTC. The numerical values of the splitting rules are derived empirically and can be found in Fig. A1. An

overview of the parameters used for the event classification is given in Tab. A1.

*Author contributions.* **S. J. Haas:** Conceptualization, formal analysis, investigation, methodology, software, visualization, writing - original draft preparation; **A. Kvas:** Conceptualization, data curation, formal analysis, investigation, methodology, software, supervision, writing - original draft preparation; **J. Fuchsberger:** Conceptualization, data curation, methodology, supervision, writing - review and editing.

*Competing interests.* The authors have no competing interests.

*Acknowledgements.* The authors thank G. Kirchengast for his support, constructive suggestions, and acquisition of funding which made this work possible. We further thank D. Scheidl for his work on the WegenerNet event type classification presented in Appendix A. This research has been supported by WegenerNet funding that is provided by the Austrian Federal Ministry for Education, Science and Research, the University of Graz, the state of Styria, and the city of Graz; detailed information can be found online (https://wegcenter.uni-graz.at/ wegenernet, last access: 2025-02-27). This research did not receive any specific grant from funding agencies in the public, commercial or

not-for-profit sectors.



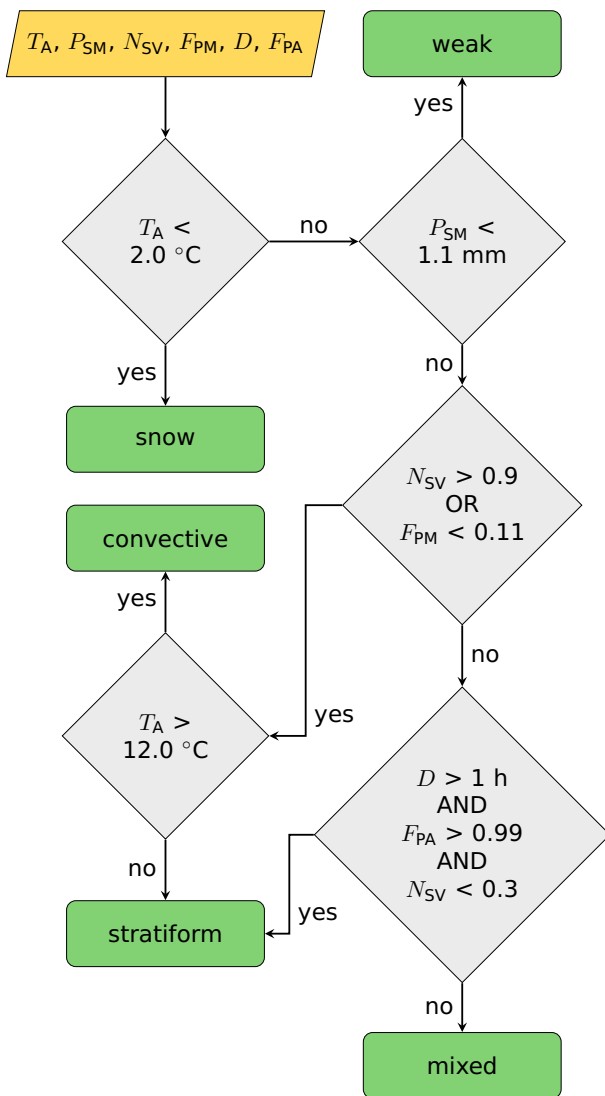

**Figure A1.** Event classification decision tree based on the features regional T2M average ($T_A$), maximum precipitation sum ($P_{SM}$), normalized spatial variability ($N_{SV}$), maximum fractional area covered by precipitation ($F_{PM}$), event duration ($D$), and fractional area at which precipitation occurred ($F_{PA}$).



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
