# Peer review of "Observation based precipitation life cycle analysis of heavy rainfall events in the southeastern Alpine forelands"

_EGUsphere, 2025_

## Referee Comment (RC2)

Review for egusphere-2025-1819: "Observation based precipitation life cycle analysis of heavy rainfall events in the southeastern Alpine forelands" by Stephanie J. Haas, Andreas Kvas, and Jürgen Fuchsberger

**General Comments**

1.) This paper nicely shows the potential of a sub-meso-scale observation network for providing insights to the initialization, evolution and decay of local Heavy Precipitation Events (HPE) over a hilly landscape in the south-east of Austria. In-situ and remotely measured meteorological variables are set into context with the precipitation development. However, I do see important factors missing in the data interpretation, which I outline in the Specific Comments below. These include the consideration of convective cold pools, advection and the synoptic situation.

2.) This paper deals with observations. However, measurement principles and errors are not discussed at all. For each observation type and variable, the authors need to provide a sound physical background, including relevant references. The resulting uncertainties should be included in the discussion of the results. This is especially true for all variables derived from the MWR and the GNSS water vapor retrievals.

3.) The MWR observations need to be interpreted in a more critical manner. Please quantify vertical resolution of the MWR retrievals and discuss the implication of vertical resolution of the MWR retrievals of the temperature and humidity profiles on the your interpretations.

**Specific Comments**

1.) Table 1: If I understand correctly, only one station provides wind speed information? If so, please indicate this in Fig. 1 and/or clarify.

2.) The observed CAPE values from the MWR in Fig. 3 are significantly smaller than the ERA5 values and also smaller than one would expect for HPE. CAPE values over 1000 J/kg are not exceptional over Europe. Please discuss where this originates from. And in this respect, describe in detail, how and what type of CAPE you have calculated. How do the values of the nearest radiosonde stations (Zagreb?) compare? Also, MSG-Seviri (and maybe already MTG?) provides a CAPE product which you should compare to.

3.) In Section 3.1 you write: *Another effect of the convective nature of HPEs is the deepening of the convective cloud system, represented by a decrease in the CBH anomaly of about 1000m prior to the event onset.* Once the air parcels have enough energy to reach the level of free convection, a deep convective system will develop and CTH will rapidly increase. Please explain in a physically plausible way, why you think the lowering of CBH is an indicator of a developing deep convective system.

4.) Fig. 4a: The vertical structure of the temperature anomaly is not discussed. I suggest analyzing the lapse rate anomalies before and after the HPE and discuss the mixing processes in the troposphere.

5.) In Section 3.1 you write: *Using the lower contour of the 80-90% RH area as a proxy for the CBH, we see a decrease in CBH of about 1 km in the 8 h before the event. This is in line with the drop in CBH anomaly already detected in Fig. 3d.* Is this really true? The strong decrease of CBH in Fig. 3 is only seen 1-2 hours before the HPE. Also,

please confirm with the actual CBH MWR/IRT retrieval that the CBH corresponds to your 80-90% contour.

6.) Fig. 5 interpretation: the increase in temperature variability goes along with the decrease of the mean temperature (Fig. 3). Relating this to clouds is probably only one part of the story. You should consider the effect of convective cold pools originating from evaporative cooling of precipitation and downward transport of upper tropospheric air (see Kirsch et al. https://doi.org/10.1002/qj.4626). Cold pools are often encountered before the actual HPE passes over the specific location. Your data are highly suited for analyzing spatial temperature variability and associated wind speeds with respect to the origin of the HPE.

7.) Fig. 7 interpretation: Please clarify in detail how you define the maximum precipitation amount? It is given in mm. What spatial and temporal extent does this amount refer to?

8.) In the discussion you write: *The energy build-up, which is also linked to the rise in temperature, is reflected by the increase of CAPE in the hours prior to the event onset (Fig. 3c).* Isn't it the airmass which is associated to a certain CAPE value and this potential energy can be set free when the surface heats or orographic lifting occurs? Wouldn't you think that the CAPE increase you see in the hours prior to the HPE is most probably due to advection of a warm and humid airmass? I suggest to check this through a more thorough characterization of the synoptic situation, e.g. by using the concept of a circulation weather type: (https://www.dwd.de/EN/research/weatherforecasting/met_applications/nwp_applications/grosswetterlagen_forecast.html)

This would provide a more comprehensive way of contextualizing your observations.

---

## Author Comment (AC2)

We thank the reviewer for their thorough assessment, feedback and comments on our manuscript. Please find our answers to the comments below with the original comment in *italics with light grey shading,* followed by our reply.

**General Comments**

1. *This paper nicely shows the potential of a sub-meso-scale observation network for providing insights to the initialization, evolution and decay of local Heavy Precipitation Events (HPE) over a hilly landscape in the south-east of Austria. In-situ and remotely measured meteorological variables are set into context with the precipitation development. However, I do see important factors missing in the data interpretation, which I outline in the Specific Comments below. These include the consideration of convective cold pools, advection and the synoptic situation.*

2. *This paper deals with observations. However, measurement principles and errors are not discussed at all. For each observation type and variable, the authors need to provide a sound physical background, including relevant references. The resulting uncertainties should be included in the discussion of the results. This is especially true for all variables derived from the MWR and the GNSS water vapor retrievals.*

   In order to provide a better overview of the measurement technique behind each variable, we expanded section 2.1. with a detailed listing and corresponding references. In addition, we added a reference to the data description preprint (Kvas et al. 2025), where details concerning quality control and inter-technique comparisons in the WEGN3D Open-Air Lab, specifically GNSS and MWR water vapor retrieval, are given.

3. *The MWR observations need to be interpreted in a more critical manner. Please quantify vertical resolution of the MWR retrievals and discuss the implication of vertical resolution of the MWR retrievals of the temperature and humidity profiles on the your interpretations.*

   Thank you for this comment. We amended the manuscript with a paragraph discussing the coarse (and varying) vertical resolution of the MWR retrievals on the temperature and humidity profiles, to put the observations in a better physical context. Specifically, we added

   **L81:** *Their main drawback is a comparatively coarse vertical resolution, especially for humidity retrievals (Barrera-Verdejo et al., 2016; Walbröl et al., 2024; Blumberg et al., 2015). Thus, MW radiometers trade vertical resolution and vertical coverage with temporal resolution compared to radiosondes (Rose et al., 2005).*

   to the dataset description in section 2.1 and

   **L212:** *The comparatively coarse vertical resolution of the radiometer-derived WEGN temperature and humidity profiles is evident, with very little vertical variation visible above 1.5 km - 2 km.*

   to our interpretation of the temperature and humdity anomaly profiles (section 3.1).

   For details concerning the impact of the vertical MWR resolution on the derived CAPE values, please see our response to your specific comment No. 2.

**Specific Comments**

1. *Table 1: If I understand correctly, only one station provides wind speed information? If so, please indicate this in Fig. 1 and/or clarify.*

   Thank you for bringing that to our attention. The station we chose to get our wind speed information from is not the only one in the WEGN3D that provides wind speed, however, it is the one that is located closely to our radiometer station. By using this station we ensure that our wind speed data is linked to the same precipitation systems as the parameters we obtain from the radiometers. To clarify this also in the manuscript we added the following sentence:

   **L98:** *For this study we decided to use the wind speed sensor that is located closest to the radiometer site to ensure that the measured wind speeds and the parameters obtained from the radiometers are connected to the same precipitation systems.*

2. *The observed CAPE values from the MWR in Fig. 3 are significantly smaller than the ERA5 values and also smaller than one would expect for HPE. CAPE values over 1000 J/kg are not exceptional over Europe. Please discuss where this originates from. And in this respect, describe in detail, how and what type of CAPE you have calculated. How do the values of the nearest radiosonde stations (Zagreb?) compare? Also, MSG-Seviri (and maybe already MTG?) provides a CAPE product which you should compare to.*

   Thank you for this observation. We attribute the lower CAPE values to the comparatively coarse resolution of the MWR-derived temperature and humidity profiles. To verify this assumption, we computed "MWR-like" profiles from Zagreb radiosonde data by adaptive smoothing to resemble the vertical resolution of the MWR (see Figure R1). The smoothing is implemented by aggregating all values within an altitude-dependent range according to Blumberg et al. 2015. We then compared surface-based CAPE values derived from the original radiosonde profiles to the "MWR-like" ones over a 4 year time span and found a CAPE reduction of 36% (see Figure R2). This behavior is also consistent with previous studies (e.g., Gartzke et al. 2017).

[Figure]

Figure R1: Comparison of original radiosonde (RS) and smoothed "microwave radiometer (MWR)-like" profiles of temperature (T), relative humidity (RH), and absolute humidity (AH) for a single launch at Zagreb 2024-07-28 12:00.

[Figure]

Figure R2: Scatter plot of surface-based CAPE values derived from radiosonde (RS) and smoothed, "mircowave radiometer (MWR)-like" profiles over a 4 year time span.

To better communicate this effect, we amended section 2.2.2. with the following paragraph:

**L129:** *Note that due to the comparatively coarse resolution of the radiometer-derived temperature and humidity profiles, the resulting CAPE values are lower than, for example, radiosonde-derived ones (e.g., Gartzke et al. 2017). We thus focus on the temporal evolution of CAPE, rather than absolute values.*

We calculate the (surface-based) CAPE using the cape_cin function (https://unidata.github.io/MetPy/latest/api/generated/metpy.calc.cape_cin.html) of the MetPy package, which follows the formula by Hobbs (1977):

$$CAPE = \int_{SL}^{LFC} R_d (T'_v - T_v)\, d\ln(p)$$

$CAPE$ … convective available potential energy

$LFC$ … pressure of level of free convection

$SL$ … pressure of surface level

$R_d$ … gas constant

$T'_v$ … parcel virtual temperature

$T_v$ … environment virtual temperature

$p$ … atmospheric pressure

We amended section 2.2.2 to make it clear that surface-based CAPE is used throughout the study.

3. *In Section 3.1 you write: Another effect of the convective nature of HPEs is the deepening of the convective cloud system, represented by a decrease in the CBH anomaly of about 1000m prior to the event onset. Once the air parcels have enough energy to reach the level of free convection, a deep convective system will develop and CTH will rapidly increase. Please explain in a physically plausible way, why you think the lowering of CBH is an indicator of a developing deep convective system.*

Thank you for pointing that out. After reading your comment and checking the development of our HPEs again, we agree that the lowering of the CBH anomaly is probably not connected to the deepening of the convective core. Since most of the investigated HPEs do not form within the 22 km x 16 km covered by the WEGN, we assume that the drop in CBH anomaly we observe stems from the displacement of no clouds/fair weather clouds with cumulus clouds that move into the study region close to a HPE. We changed the corresponding sentences in the manuscript and replaced the term 'deepening of the convective system' with 'arrival of the convective system'. In addition we changed the following sentences:

**L140:** *Another effect of the convective nature of HPEs is the deepening of the convective cloud system, represented by a decrease in the CBH anomaly of about 1000 m prior to the event onset.*

[changed to]

**L169**: *Most HPEs observed do not form within the comparatively small region of the WEGN. This means that before a HPE the sky is either clear or filled with some fair weather clouds which get displaced by cumulus clouds close to the actual event. This is represented by a decrease in the CBH anomaly of about 1000 m prior to the event onset.*

4. *Fig. 4a: The vertical structure of the temperature anomaly is not discussed. I suggest analyzing the lapse rate anomalies before and after the HPE and discuss the mixing processes in the troposphere.*

Thank you for this interesting suggestion. We investigated the LR and added a plot similar to Fig. 4 in the appendix. We also added a few sentences describing the findings of this analysis to the main paper:

**L184:** *As an indicator for the atmosphere's stability, we additionally investigate the environmental lapse rate (LR), shown in Appendix B, Fig. B1. Before the event onset, we observe high LR values that point towards an unstable, and hence a thunderstorm favoring, atmosphere (Daidzic, 2019). In the hours after the event, we observe decreased LR values. At lower altitudes, the difference between pre and after event LRs is highest, which we attribute to the increased moisture at these levels. At higher altitudes the decrease in LR values might be connected to the release of latent heat during the rainfall event.*

Concerning the suggestion to discuss the mixing processes in the troposphere we consider this beyond the scope of our observation-based study.

5. *In Section 3.1 you write: Using the lower contour of the 80-90% RH area as a proxy for the CBH, we see a decrease in CBH of about 1 km in the 8 h before the event. This is in line with the drop in CBH anomaly already detected in Fig. 3d. Is this really true? The strong decrease of CBH in Fig. 3 is only seen 1-2 hours before the HPE. Also, please confirm with the actual CBH MWR/IRT retrieval that the CBH corresponds to your 80-90% contour.*

During the review we found a bug in our plot script. Accidentally, not the 8h before the event were shown in the vertical plots but only about 1.5 hours. After comparing the actual CBH to the 80 % RH contour (see Figure R1 below), we find that our findings remain the same. Even though the CBH is located a few meters below the 80 % contour line, both the 80 % RH contour and the CBH decrease in the hours before the event onset. The strongest decrease in the 80 % RH contour line sets in ~2 h before the event onset which is in-line with the decrease in CBH anomaly shown in Figure 3 of the manuscript. To clarify this in the manuscript we changed the corresponding line to:

**L157:** *Using the lower contour of the 80-90 % RH area as a proxy for the CBH, we see a decrease in CBH of about 1 km in the 8 h before the event. This is in line with the drop in CBH anomaly already detected in Fig. 3d.*

[changed to]

**L193:** *Using the lower contour of the 80-90 % RH area as a proxy for the CBH, we see a decrease in CBH of about 2 km in the 2 hours before the event. This roughly corresponds to the drop in CBH anomaly already detected in Fig. 3d.*

[Figure]

Figure R3: Median vertical structure of relative humidity for WEGN3D (as shown in panel b of Figure 4 in the main paper). The yellow line indicates the 80% RH contour line, the dotted white line marks the CBH.

6. *Fig. 5 interpretation: the increase in temperature variability goes along with the decrease of the mean temperature (Fig. 3). Relating this to clouds is probably only one part of the story. You should consider the effect of convective cold pools originating from evaporative cooling of precipitation and downward transport of upper tropospheric air (see Kirsch et al. https://doi.org/10.1002/qj.4626). Cold pools are often encountered before the actual HPE passes over the specific location. Your data are highly suited for analyzing spatial temperature variability and associated wind speeds with respect to the origin of the HPE.*

Thank you for this interesting suggestion. After reading your comment, we checked the HPEs for cold pool occurrences. Though we do find cold pools prior to the events for some of the HPEs (Figure R2), we have the reoccurring issue that most of our events do not form within the comparatively small region covered by the WEGN. To acknowledge that the observed temperature variability does not solely stem from cloud formation, we added the following sentence to our manuscript:

**L221:** *Another reason for the increase in temperature variability might come from convective cold pools (Kirsch et al., 2024) which are often detected at the location of HPEs before the event onsets. While we do find indications of such cold pools for a few of the investigated HPEs (not shown), most of the HPEs do not form directly in the region covered by the WEGN, which means that potential cold pools cannot be found in that area as well.*

[Figure]

Figure R2: Event precipitation amount (left panel) and temperature anomaly 1 h prior to the onset of the 2021-07-30 HPE. Areas with temperature anomalies <= -2°C (i.e. possible cold pools) are marked with the black contour lines in the right panel.

7. *Fig. 7 interpretation: Please clarify in detail how you define the maximum precipitation amount? It is given in mm. What spatial and temporal extent does this amount refer to?*

The maximum precipitation amounts shown in Figure 7 are the maximum 5 min amounts recorded during the event, which correspond to an individual station. Meaning that for each time step during the event the 5 min precipitation amount of the station with the highest value is selected. To state this more clearly in the manuscript, we added this information in the following sentence:

**L199:** *Figure 7 depicts the development of the maximum precipitation amount (PAx), 2 m air temperature anomaly, and IWV anomaly during the event, as well as, the spatial variability of these parameters.*

[changed to]

**L238:** *Figure 7 depicts the development of the maximum 5 min precipitation amount (PAx), 2 m air temperature anomaly, and IWV anomaly during the event, as well as, the spatial variability of these parameters.*

8. *In the discussion you write: The energy build-up, which is also linked to the rise in temperature, is reflected by the increase of CAPE in the hours prior to the event onset (Fig. 3c). Isn't it the airmass which is associated to a certain CAPE value and this potential energy can be set free when the surface heats or orographic lifting occurs? Wouldn't you think that the CAPE increase you see in the hours prior to the HPE is most probably due to advection of a warm and humid airmass? I suggest to check this through a more thorough characterization of the synoptic situation, e.g. by using the concept of a circulation weather type: (https://www.dwd.de/EN/research/weatherforecasting/met_applications/ nwp_applications/grosswetterlagen_forecast.html) This would provide a more comprehensive way of contextualizing your observations.*

Thank you for that input. We agree that the observed rise in CAPE is probably not directly linked to the HPEs themselves. As already mentioned in our answer to specific comment #3, most of the investigated HPEs do not actually form within the

region of the WEGN3D and the CAPE values observed by the WEGN3D are therefore not representative for the events themselves.

To emphasize this also in the manuscript, we changed the corresponding sentence to:

**L278:** *The energy build-up, which co-occurs with a rise in temperature, is reflected by the increase of CAPE in the hours prior to the event onset (Fig.3c).*

We followed your suggestion of checking the weather types and found that many of our events occurred on days with the BM (*"Hochdruckbrücke Mitteleuropa"*) synoptic situation (see Fig. R3), which is characterized by a band/area of high pressure over Central Europe. This is a very curious finding, since the HB situation is not very common over the year and only ~7% of all days exhibit this synoptic condition:

[https://www.pik-potsdam.de/en/output/publications/pikreports/.files/pr119.pdf](https://www.pik-potsdam.de/en/output/publications/pikreports/.files/pr119.pdf)

Though we find this topic highly interesting, we consider it beyond the scope of this study, where we focus on the observation-based life-cycle of rainfall events. However, we are currently in the process of planning further studies that focus on the drivers of HPEs on different scales, where we will also investigate the circulation patterns connected to these events with the amount of attention and detail required by this topic.

[Figure]

Figure R3: Number of Großwetterlagen (GWL) connected to the HPEs investigated in the study.

**REFERENCES**

Gartzke, J., R. Knuteson, G. Przybyl, S. Ackerman, and H. Revercomb, 2017: Comparison of Satellite-, Model-, and Radiosonde-Derived Convective Available Potential Energy in the Southern Great Plains Region. *J. Appl. Meteor. Climatol.*, **56**, 1499–1513, https://doi.org/10.1175/JAMC-D-16-0267.1.

Hobbs, P. V., and J. M. Wallace, 1977: *Atmospheric Science: An Introductory Survey.* Academic Press, 350 pp.

Kvas, A., Kirchengast, G., and Fuchsberger, J.: High-resolution atmospheric data cubes from the WegenerNet 3D Open-Air Laboratory for Climate Change Research, Earth Syst. Sci. Data Discuss. [preprint], https://doi.org/10.5194/essd-2025-176, in review, 2025.

Blumberg, W. G., Turner, D. D., Löhnert, U., & Castleberry, S. (2015). Ground-Based Temperature and Humidity Profiling Using Spectral Infrared and Microwave Observations. Part II: Actual Retrieval Performance in Clear-Sky and Cloudy Conditions. *Journal of Applied Meteorology and Climatology*, *54*(11). https://doi.org/10.1175/JAMC-D-15-0005.1

---

## Referee Report (RR1)

Second review for egusphere-2025-1819: "Observation based precipitation life cycle analysis of heavy rainfall events in the southeastern Alpine forelands" by Stephanie J. Haas, Andreas Kvas, and Jürgen Fuchsberger

The authors have nicely addressed most of my major points raised. I suggest further adjustments to the following open issues, from my point of view.

1.) Instrument uncertainties: The MWR is an essential instrument in your study. And you are using it to analyze the atmosphere around heavy precipitation events. Ground-based MWR are prone to errors due to a wet radome. Please provide information on how you quality control/assure your observational data in this sense. You may find some useful information here: https://zenodo.org/records/11422901 or here: https://egusphere.copernicus.org/preprints/2025/egusphere-2025-1727/

2.) In section 3.1 you now write: *Most HPEs observed do not form within the comparatively small region of the WEGN. This means that before a HPE the sky is either clear or filled with some fair weather clouds which get displaced by cumulus clouds close to the actual event. This is represented by a decrease in the CBH anomaly of about 1000 m prior to the event onset.*

You are implying a physical causality here that I cannot recognize. 1h before the HPE, you see an >1kg/m2 increase in IWV (Fig. 3). Don't you think this could lead to lowering of the CBH, e.g. through a lowering of the LCL or CCL? Did you check the near-surface spread (T-Td) for this?

3.) It great that you have provided a figure with the LR time series. But why put it in the appendix? Your paper does not have too many figures and the LR figure is an additional analysis, so I suggest moving it to the main body of the paper. In any case, you need to discuss this figure more quantitively if you add it to the paper. Fig. B1 before the HPE makes sense (strong instability close to the surface and then conditional stability above). But I wonder what is going on after the HPE. Why do you see a persistent inversion up to 12h after the event peaking around 500m?

4.) Cold pools, you now write: *Another reason for the increase in temperature variability might come from convective cold pools (Kirsch et al., 2024) which are often detected at the location of HPEs before the event onsets. While we do find indications of such cold pools for a few of the investigated HPEs (not shown), most of the HPEs do not form directly in the region covered by the WEGN, which means that potential cold pools cannot be found in that area as well.*

Could you include an argument why cold pools cannot be detected if the HPE doesn't form in the WEGN region? Cold pools are generally defined by rapid horizontal wind increase (gust front) and simultaneous temperature drop shortly before the event, both which you nicely see in Fig. 3.

---

## Author Response (AR2)

We thank the editor and the reviewer for their thorough assessment, feedback and comments on our manuscript. Please find our answers to the comments below with the original comment in *italics with light grey shading,* followed by our reply.

**COMMENTS BY THE REVIEWER**

*1.) Instrument uncertainties: The MWR is an essential instrument in your study. And you are using it to analyze the atmosphere around heavy precipitation events. Ground-based MWR are prone to errors due to a wet radome. Please provide information on how you quality control/assure your observational data in this sense. You may find some useful information here: https://zenodo.org/records/11422901*
*or here: https://egusphere.copernicus.org/preprints/2025/egusphere-2025-1727/*

Thank you for providing the link to the preprint, we are aware of the methodology by the authors and have already adopted it for the next iteration of our quality control system. For the MWR data used in this study (Version 1.0), we use a rather conservative radome time-to-dry estimate of 30 minutes (see Kvas et al. 2024 for details) after the last recorded rainfall (as measured by the radiometer-mounted weather stations and neighboring rain gauges) at the radiometer site. In contrast, Böck et al. (2025) observe time-to-dry values of 20 minutes as a worst case (Fig. 12). This means that generally more observations than necessary are flagged, but ensures that a wet radome does not influence our conclusions. We amended the manuscript with:

> L240: To avoid these biases influencing our analysis, we exclude all microwave radiometer observations between the first recorded rainfall and 30 min after the last recorded rainfall at the radiometer site.

to clarify our approach.

Böck, T., Löffler, M., Marke, T., Pospichal, B., Knist, C., and Löhnert, U.: Instrument uncertainties of network-suitable ground-based microwave radiometers: overview, quantification, and mitigation strategies, EGUsphere [preprint], https://doi.org/10.5194/egusphere-2025-1727, 2025.

Kvas, A., Fuchsberger, J., & Kirchengast, G. (2024). Algorithm Theoretical Basis Document for Version 1.0 of the WegenerNet 3D Observing System L1 and L2 Data Products (WegenerNet Technical Report No. 2/2024). Wegener Center for Climate and Global Change, University of Graz.
https://wegenernet.org/downloads/Kvas_et_al_2024_WEGN3D_v1_ATBD-WEGN-TR-2-2024.pdf

*2.) In section 3.1 you now write: Most HPEs observed do not form within the comparatively small region of the WEGN. This means that before a HPE the sky is either clear or filled with some fair weather clouds which get displaced by cumulus clouds close to the actual event. This is represented by a decrease in the CBH anomaly of about 1000 m prior to the event onset.*

*You are implying a physical causality here that I cannot recognize. 1h before the HPE, you see an >1kg/m2 increase in IWV (Fig. 3). Don't you think this could lead to lowering of the CBH, e.g. through a lowering of the LCL or CCL? Did you check the near-surface spread (T-Td) for this?*

After reading your comment, we checked the T-Td vertical structure and got very similar results to the vertical RH structure shown in Figure 4b. In our opinion, there are two possible conditions before an HPE in our study region:

1. The HPE forms outside the WEGN region and moves into the study region. This is the situation we described in the manuscript, where there are no clouds (or some fair-weather clouds) in the study region before the event, which then get displaced by cumulus clouds. Since the cumulus clouds have a lower CBH than the fair-weather clouds, this would explain the observed drop of CBH.

2. The HPE forms inside the WEGN region. In this case we agree, that the lowering of the CBH is probably connected to a lowering in LCL.

Considering that we know that most of the observed events do not form within the region of the WEGN, we cannot be sure which is the dominating mechanism that leads to the observed drop in CBH. We changed the corresponding sentences in the manuscript to clarify that our explanation might not be the only one:

*L171: There are multiple mechanisms that can explain the decrease in CBH anomaly of about 1000 m detected close to the event onset. Connected to the increase in IWV anomaly described above, a lowering of the lifting condensation level could result in a lowering of the CBH. However, we also find that most HPEs observed do not form within the comparatively small region of the WEGN. This means that before a HPE the sky is either clear or filled with some fair-weather clouds which get displaced by cumulus clouds close to the actual event, which could also lead to the observed decrease in CBH.*

*3.) It great that you have provided a figure with the LR time series. But why put it in the appendix? Your paper does not have too many figures and the LR figure is an additional analysis, so I suggest moving it to the main body of the paper. In any case, you need to discuss this figure more quantitively if you add it to the paper. Fig. B1 before the HPE makes sense (strong instability close to the surface and then conditional stability above). But I wonder what is going on after the HPE. Why do you see a persistent inversion up to 12h after the event peaking around 500m?*

Thank you for that input. We consider the analysis of the LR as supporting information and therefore decided to leave it in the appendix. There are two possible reasons for the observed inversion up to 12 h after the event:

1. The difference between pre- and post-event temperature is higher at lower altitudes (Fig. 4a). This means that the lower layers are cooler after the event than the higher layers which leads to an inversion in the LR.

2. As shown in Figure 2, most of our events occur in the afternoon (around 15 h). The 12 h after the event are therefore mainly during the night time, when inversion layers often occur because the surface-near layers are cooling faster than the higher layers. After the 12 h, the next day already started and temperatures begin to rise again, and the inversion layer vanishes.

We agree that our description of the LR figure was a bit scarce. We therefore adjusted the following paragraph:

*L191: In the hours after the event, we observe decreased LR values and a persistent inversion layer at roughly 500 m. Figure 4a shows that the temperature difference between pre and after event is higher at lower altitudes, which we attribute to the increased moisture at these levels. As a consequence, the lower layers are cooler after the event then higher layers, resulting in an inversion layer. At higher altitudes the decrease in LR values might be connected to the release of latent heat during the rainfall event. Further, due to the events' tendency to occur mainly in the afternoon, the hours after the event are often during the night time, where radiation effects often lead to LR inversions.*

*4.) Cold pools, you now write: Another reason for the increase in temperature variability might come from convective cold pools (Kirsch et al., 2024) which are often detected at the location of HPEs before the event onsets. While we do find indications of such cold pools for a few of the investigated HPEs (not shown), most of the HPEs do not form directly in the region covered by the WEGN, which means that potential cold pools cannot be found in that area as well.*

*Could you include an argument why cold pools cannot be detected if the HPE doesn't form in the WEGN region? Cold pools are generally defined by rapid horizontal wind increase (gust front) and simultaneous temperature drop shortly before the event, both which you nicely see in Fig. 3.*

You are correct, we do see an increase in wind speeds and a drop in temperatures shortly before the event onset in our data. However, since we know that most of our events do not form within our study region, cold pools connected to these events would likely also not be located in our comparatively small study area. We therefore do not want to argue that the rise in wind speeds and the drop in temperature we observe in our region is connected to the formation of cold pools. Such a claim would need further investigation, which we consider to be beyond the scope of this study.

**COMMENTS BY THE EDITOR**

*L10: The sentence "Beginning with ..." seems to me grammatically wrong, please rephrase.*

Thank you for pointing that out. We changed the corresponding sentence to:

*L11: Beginning with the event formation stage (i.e., the 8 hours before the event), temperatures are usually already quite high and continue to rise, while the first clouds begin to form, followed by an increase in wind speeds and a darkening sky.*

*L13: please specify "we find an increase ...": when? during the 8 h before the event? or do you mean that during events these parameters are increased compared to climatology? please clarify.*

To clarify our statement, we added *"… in the 8 h prior to the event onset."* at the end of the sentence.

*abstract: I would find it very interesting if you could mention in the abstract the timing of your 94 events (at what time of the day do they typically occur? and in what months of the year?).*

To characterize our events more precisely, we adjusted the following sentence:

*Here we follow the life cycle of 94 heavy rainfall events by investigating multiple atmospheric parameters in WEGN3D and global reanalysis data.*

[changed to]

*L9: Here we follow the life cycle of 94 heavy rainfall events, which mainly occur in the afternoon hours in the warm season, by investigating multiple atmospheric parameters in WEGN3D and global reanalysis data.*

*L36 but also in other places: please list references chronologically*

Thank you for pointing this out to us, we changed ordered the references accordingly.

*L68: "unprecedented holistic way" is maybe a bit exaggerated, maybe chose a more modest formulation.*

We changed the corresponding sentence to:

*L68: In this study, we leverage the different instruments available in the WEGN3D Open-Air Lab to holistically investigate local HPE precursors in high-resolution data.*

*Eq. 1: this is not good notation, why is hourly precipitation called I_P and 5-min precipitation PA? I would just write that hourly precipitation rates are estimated from 5-min accumulated measurements by multiplying with a factor of 12.*

Thank you for noticing. We removed the equation, since it is not used later on anyways and changed the corresponding sentence to:

*L117: Note that due to the 5 min temporal resolution of the WEGN, hourly precipitation rates are derived by multiplying the 5 min precipitation amounts with a factor of 12.*

*Section 2.2.2: please re-consider your notation. I have never seen AH for absolute humidity (you mean specific humidity in g/kg? or mixing ratio in g/m3?); q or w would be the more common symbols. For all variables, please indicate their unit, and if you use symbols (like T for temperature, then T should be in italics). Abbreviations (RH, IWV, ...) should remain in roman letters.*

In our study we use the absolute humidity (i.e., the mass of water vapor per volume of air) and not the specific humidity. Analogue to RH for relative humidity, the abbreviation AH is not uncommon in climate studies. See also the corresponding Wikipedia article: https://en.wikipedia.org/wiki/Humidity#Absolute_humidity. We therefore decided to stick to AH for absolute humidity.

We changed all occurrences of T to italics. The units of all parameters can be found in Table 1.

*Figure 6: it is unclear to me what you mean by "regional anomaly" - is this a deviation from the domain mean, or from climatology? And which event are you showing in Fig. 6 (indicate the date)? And what do you mean by "accumulated PA" (is this the total precipitation within 20 minutes)?*

After reading your comment, we agree that the figure was not described sufficiently. We added a short description of how the regional anomaly is computed and some additional information about the event date and the meaning of the accumulated PA in the manuscript where the figure is introduced. We also added the event date in the figure caption.

*L242: Figure 6 shows a snapshot of the regional T2M anomaly, along with accumulated precipitation amounts (from event start to 20 min into the event) derived from radar and gauge measurements, 20 min into the event on 2022-09-15. The T2M anomaly is calculated by subtracting the spatial mean over the whole region from the data.*

*Figure 7: for panel c, the caption says it is T2M, but it is likely a T2M anomaly (again, relative to what?)*

Thank you for bringing that to our attention. We corrected the figure caption. A description of how the anomaly is calculated is given in Section 2.2.2. To emphasize that the anomaly shown in Figure 7 is calculated in the same way, we added a sentence referring to the corresponding section:

*L249: The temperature anomaly is again calculated with the steps described in Section 2.2.2.*

*L317: again, it is important that the reader here understand what you mean by "anomaly" (deviation from climatology? from domain mean? From ...?)*

To clarify what we mean by "anomalies", we added a sentence in Section 2.2.2 where the preprocessing of the data is explained:

*L141: Anomalies shown in this study should therefore be considered as 'event anomalies'.*

*L326: it is good tradition when reporting about observations, to make the point that they are important for improving models (and predictions). But do you have a specific suggestion how your results could help with this? I very much like your observational analysis, but I don't see immediately how a model can be improved given your study, and therefore the final statement sounds a bit overly general. Maybe you would like to emphasise that one could do similar analyses with high-resolution weather prediction models and quantify whether they have the correct time evolution of T2M, CAPE etc. prior to HPEs?*

Thank you for this suggestion. As we already cover this topic in the Discussion, we added the following sentences to the Discussion Section:

*L310: Another potential application is the verification of high-resolution weather models by comparing the temporal evolution of the parameters examined in this study (cf. Table 1) with the corresponding model outputs. The models could also be tuned to more accurately reproduce the observed behavior of these parameters.*